# Multiple-Cycle Polymeric Extracellular Vesicle Precipitation and Its Evaluation by Targeted Mass Spectrometry

**DOI:** 10.3390/ijms22094311

**Published:** 2021-04-21

**Authors:** Jisook Park, Eun-Bi Go, Ji Sun Oh, Jong Kyun Lee, Soo-Youn Lee

**Affiliations:** 1Samsung Biomedical Research Institute, Samsung Medical Center, Sungkyunkwan University School of Medicine, Seoul 06351, Korea; js944837@skku.edu; 2Samsung Biomedical Research Institute, Samsung Medical Center, Seoul 06351, Korea; geb926@naver.com (E.-B.G.); jisun.oh@daum.net (J.S.O.); 3Division of Gastroenterology, Department of Medicine, Samsung Medical Center, Sungkyunkwan University School of Medicine, Seoul 06351, Korea; jongk.lee@samsung.com; 4Department of Laboratory Medicine and Genetics, Samsung Medical Center, Sungkyunkwan University School of Medicine, Seoul 06351, Korea; 5Department of Clinical Pharmacology and Therapeutics, Samsung Medical Center, Seoul 06351, Korea; 6Department of Health Science and Technology, Samsung Advanced Institute of Health Science and Technology, Sungkyunkwan University, Seoul 06351, Korea

**Keywords:** extracellular vesicle, polymeric precipitation, multiple reaction monitoring, pancreatic cancer

## Abstract

The multiple roles of extracellular vesicles (EVs) in pathogenesis have received much attention, as they are valuable as diagnostic and prognostic biomarkers. We employed polymeric EV precipitation to isolate EVs from clinical specimens to overcome the limitations of ultracentrifugation (UC), such as low protein yields, a large volume of specimens used, and time requirements. Multiple-cycle polymeric EV precipitation was applied to enhance the purity of the EV fractions with a small sample volume. Then, the purity was assessed using a multiple reaction monitoring (MRM) panel consisting of alpha-2-macroglobulin (A2M), thrombospondin 1 (THBS 1), galectin 3 binding protein (LGALS3BP), and serum albumin (ALB). For purity evaluation, the EV fractions isolated from blood specimens were subjected to shotgun proteomics and MRM-based targeted proteomics analyses. We demonstrate that the modified method is an easy and convenient method compared with UC. In the shotgun proteomics analysis, the proteome profile of EV fraction contains 89% EV-related proteins by matching with the EVpedia database. In conclusion, we suggest that multiple-cycle polymeric EV precipitation is not only a more effective method for EV isolation for further proteomics-based experiments, but may also be useful for further clinical applications due to the higher EV yield and low sample requirements.

## 1. Introduction

Extracellular vesicles (EVs) are spherical cell-derived membrane particles classified into exosomes (30–120 nm), microvesicles (100–1000 nm), and apoptotic bodies (800–5000 nm) according to cellular origin, function, and biogenesis [1,2]. EVs contain bioactive molecules such as DNA, mRNA, microRNA(miRNA), proteins, and lipids, which are released by most cell types into body fluids (e.g., serum, urine, and cerebrospinal fluids) [3].

EVs are intracellular hormone-like messengers such as protein carriers or RNA carriers that are reported to be deeply involved in diseases and biological processes such as immune response regulation, blood coagulation, cell migration, cell differentiation, and cell-to-cell communication [4,5,6,7,8]. In particular, these vesicles are highly released in diseases including cancer and play a role in spreading proteins that cause disease to other cells [9]. Therefore, they have attracted much attention due to their roles under physiological and pathological conditions as diagnostic biomarkers and therapeutic targets [10,11]. In particular, the EV proteome profile has been highlighted as it is expected to be useful for diagnostic biomarkers and therapeutic targets [12,13,14,15,16].

Differential ultracentrifugation (UC) is the most traditional protocol used to isolate EV from various body fluids to date [17]. However, there are several limitations to its clinical application including the low yield of EVs, large volume of specimens, and above all, its time-consuming process. Polymer-based precipitation is more advantageous for clinical applications than UC because of the small sample volume, easy protocols, and short turnaround time [18,19]. However, contamination of non-EV proteins has been an obstacle to increasing the contents of EV proteins in polymeric methods [20]. The EVs acquired from biological fluids inevitably include high levels of serum-derived proteins, which are a major impediment to EV proteomics. UC is still considered the gold standard for EV isolation, especially for proteomics experiments.

Recent studies showed that multiple cycles of UC provided higher purity of EVs than classical UC purification [21]. Unfortunately, with multiple-cycle UC, a large sample volume is needed for EV isolation, whereas a smaller amount of EVs is recovered. Therefore, it is an urgent requirement to improve EV purification technology for further clinical applications.

A large sample size, poor yield of EVs, and the time-consuming process are major problems in the clinical application of EV biomarkers. The evaluation of impurities from the EV purification process is critical for downstream EV proteomics analysis. Evaluation of purity is limited to measurement of serum albumin using Western blotting without EV-specific proteins [22]. Development of an analysis method for purity evaluation is necessary. For quantitative analysis of newly discovered proteins, multiple reaction monitoring (MRM) is a more useful technique than other antibody-based methodologies (e.g., Western blotting or enzyme-linked immunosorbent assay) due to its rapid development and clinical application.

Polymeric reagents reduce the solubility of EVs and allow less-soluble components to precipitate out of the solution. This method produces a high yield of EVs, but contamination of non-EVs proteins is a major problem. We expected that repeated treatment of polymeric reagents can wash away contaminated proteins and increase purity. Therefore, we constructed a modified method that includes multi-cycle processing to enhance the purity.

Here, we present multiple-cycle polymeric EV precipitation with high purity, which may be applicable in clinical practice and provides EV marker proteins for purity evaluation. This is the first report on the usefulness of multi-cycle polymer-based reagent treatment compared with multi-cycle UC [23], including quantitative results of each EV fraction acquired from a different reagent processing cycle and verified by MRM assay.

## 2. Results and Discussion

### 2.1. The Size Distribution and Morphology Analysis of EVs

The EVs isolated from human serum by UC and multiple-cycle polymeric EV precipitation were characterized by TEM and a Zetasizer Nano ZS. The TEM and Zetasizer analyses were applied to confirm the size distribution and morphology of the isolated EVs (Figure 1). The average sizes of the UC and four-cycle EV fractions were 114 nm (50–160 nm) and 40 nm (25–80 nm), respectively. The four-cycle EV fractions were a little smaller than the UC fractions.

### 2.2. Identification of EV Proteins Using Proteomic Analysis

A total of 612 proteins (unique peptides ≥2) was identified in the four-cycle EV fractions from human serum by shotgun proteomic analysis. Of them, 539 proteins (89%) were included in the EVpedia depository (http://evpedia.info, accessed on 18 May 2019), a public database for extracellular vesicle research. These proteins were highly enriched in extracellular exosomes, extracellular vesicles, and extracellular regions by functional enrichment analysis using FunRich version 3.1.3 (Figure 2). We could not prepare enough EVs fractions from UC experiments due to the limitations due to sample volume and very low recovery rate. Therefore, our current data were compared with the proteome profile from another investigation using multiple-cycle UC [21]. In brief, Kim et al. reported that 478 proteins including a single unique peptide were identified in LC-MS/MS analysis of EV proteins enriched from 4 mL of serum using five cycles of UC. Among these proteins, we extracted 273 having two or more unique peptides to compare with our identified proteins.

A total of 139 proteins was commonly identified in both the UC fraction and our four-cycle EV fraction, and 465 proteins were exclusively included in the four-cycle EV fraction. More plasma membrane proteins were enriched in the four-cycle EV fraction (99 proteins) than in the UC fraction (38 proteins) (Appendix A), and 27 clusters of differentiation (CD) proteins were identified in both fractions (Table 1). As a result, our data indicate that EV isolation by our protocol can be applied to various proteome-based investigations.

### 2.3. EV MRM Assay Development

A total of 234 MRM transitions were generated against the 26 proteins and 78 peptides common among the shotgun data and top 100 EV proteins obtained using Skyline software. The MRM methods were experimentally refined with the EV fractions obtained from pooled serum. Of them, MRM assays of six proteins were finally established (Appendix A and Appendix A). We measured the levels of these proteins in the EV fractions (one to four cycles) obtained from pooled sera, of which three proteins, A2M, THBS1, and LGALS3BP, increased significantly after EV purification (Appendix A). Therefore, these three proteins were chosen as EV marker proteins for the EV purity evaluation (Table 2).

### 2.4. Purity Evaluation by Immunoblotting and MRM Assay

Western blot analysis of CD9, CD63, and CD81 was performed to assess the losses and concentration efficiency of EV purification. In the current study, there was no significant intensity difference in the band of each protein between UC and 1–4-cycle EV fractions when an equal amount of the EV fraction was applied (Figure 3A). Meanwhile, its total protein amount was at least 4.6-fold higher than that of UC, even though the volume of the sample we treated was 16 times smaller (data not shown).

Serum albumin (ALB) was used to assess whether serum-derived contaminant proteins were removed or not because it is the most abundant component in the blood sample. We measured the ALB concentrations in the 1–4-cycle EV fractions and UC fraction obtained from the pooled sera. One treatment of polymeric reagents was insufficient for the removal of serum-derived contaminants. Approximately 30% of ALB was removed in the one-cycle EV fraction, 93% after two cycles, 99% with four cycles, and 98% for the UC fraction (four-cycle) compared with the non-purified fraction (Figure 3B). To verify the above results, we performed an MRM assay of ALB and three EV proteins of the two-cycle EV fractions and non-purified fractions obtained from 20 serum samples (Figure 3C). Consistent with previous MRM results, these proteins (A2M, THBS1, and LGALS3BP) increased significantly after EV purification, while the ALB levels were dramatically reduced up to 99% after EV purification (*p*-value < 0.0001). Interestingly, LGALS3BP was not detectable in the non-purified fraction but was 7.3-fold enriched after EV purification.

As a result, our data demonstrate that multiple-cycle polymeric EV precipitation is excellent at removing serum-derived contaminant proteins and enriching EV-related proteins. Therefore, we expect the MRM panel consisting of three EV proteins and ALB to be useful for assessing EV purity isolated by various techniques.

## 3. Experimental Section

### 3.1. Clinical Samples

This study was approved by the Institutional Review Board (IRB) of Samsung Medical Center (Seoul, Korea) (IRB file No. 2008-06-007-026). We recruited 10 healthy controls (HC) and 10 patients with pancreatic cancer (PC) for EV purity assessment and LC-MS/MS EV fraction experiments. All serum specimens were obtained by centrifugation of blood at 2330× *g* for 5 min and stored at −70 °C for further analysis.

### 3.2. EV Purification

The EVs isolation and characterization were performed according to the exosome guidelines [24]. EV fractions were acquired from UC and a multiple-cycle polymeric EV precipitation method. For the UC fractions, 4 mL of pooled human serum was centrifuged at 10,000× *g* for 30 min, and the supernatant was filtered through a 0.2 μm PVDF filter (Agilent Technologies, Santa Clara, CA, USA). The samples were mixed with 6 mL of PBS and then ultracentrifuged at 100,000× *g* for 120 min at 4 °C (four times). After the UC step, the supernatant was suctioned, and pellets were reconstituted with 100 μL of PBS.

For polymeric EV precipitation, ExoQuick reagent (System Biosciences Inc., Palo Alto, CA, USA) was used according to the manufacturer’s recommendations. Briefly, the collected serum was centrifuged at 3000× *g* for 15 min to remove cellular debris. We mixed 25 μL of ExoQuick reagent and 100 μL of the supernatant were mixed and incubated for 30 min, followed by centrifugation at 1500× *g* for 30 min at 4 °C. The supernatant was discarded, and the remaining material was centrifuged once more (1500× *g* for 5 min) to remove residual fluid. This step was repeated up to 4 times. The final pellets were reconstituted in 100 μL of PBS (Figure 4).

### 3.3. Dynamic Light Scattering (DLS) Technology

DLS measurements were performed using a Zetasizer Nano ZS (Malvern Instruments, Malvern, UK) to determine the particle size distribution of the EVs isolated from different methods. The isolated EVs were diluted 1000-fold with deionized water and tested using a Zetasizer Nano ZS according to the manufacturer’s instructions.

### 3.4. Transmission Electron Microscopy (TEM)

We fixed 5 μL of freshly prepared EVs with 5 μL of 4% paraformaldehyde for 5 min at room temperature. About 10 μL of the mixture was dropped onto carbon-coated copper grids and then air-dried for 40 min. After washing with PBS, the sample was fixed with 2.5% glutaraldehyde for 5 min and stained with UranyLess (Electron Microscopy Sciences, Hatfield, PA, USA) for 2 min, followed by washing in distilled water. The sample was dried and subjected to transmission electron microscopy (TEM) analysis.

### 3.5. Western Blot Analysis

The protein lysate of the isolated EVs was separated on 4–12% NuPAGEL Bis-Tris gel (Invitrogen, Carlsbad, CA, USA) and transferred to an Immun-Blot PVDF membrane (Bio-Rad, Hercules, CA, USA). The membranes were blocked in 3% BSA in phosphate-buffered saline (PBS) overnight at 4 °C and then incubated with primary antibodies: anti-CD9 (1:1000; Abcam, Cambridge, UK), anti-CD63 (1:1000; Abcam, Cambridge, UK), and anti-CD81 (1:1000; Abcam, Cambridge, UK) for 2 h at room temperature. The membranes were washed three times with PBS, incubated with the corresponding secondary antibody, anti-rabbit (1:3000), for 1 h at room temperature, and washed with PBST. Signals were visualized after incubation using a Pierce ECL (Thermo, Rockford, IL, USA) and a LAS4000 Imaging System (Fujifilm Life Science, Tokyo, Japan).

### 3.6. Shotgun Proteomics Analysis

A total 500 μg of purified EV fractions obtained from pooled sera were subjected to in-solution digestion with trypsin followed by nano-LC-MS/MS analysis. The peptide mixtures were fractionated by reverse phase HPLC (4.6 × 150 mm ID, 2.5 μm particle) into 42 fractions. Then, the 42 fractions were combined into 10 fractions and analyzed using an Orbitrap Fusion Lumos mass spectrometer (Thermo Finnigan, San Jose, CA, USA) equipped with a nano-electrospray ion source.

The peptide mixtures were loaded onto a trap column (Thermo Scientific EASY-Spray LC Columns 160 nL enrichment column and 75 μm × 50 cm packed C18, 2 μm particles, 100 A) connected to the Orbitrap Fusion Lumos mass spectrometer at a flow rate of 4 μL/min. The mobile phase consisted of buffer A (0.1% formic acid in water) and buffer B (0.1% formic acid in ACN). After injecting a sample onto the column, a 120 min gradient method was used to separate the peptide mixture. Sample loading onto the analytical column was performed at 3% buffer B, the mobile phase was held at 4% buffer B for 1 min, followed by a linear gradient to 32% buffer B over 91 min and then a linear gradient to 80% buffer B over 8 min at a flow rate of 300 nL/min.

All MS/MS spectra were searched against Uniprot Human 2018.1.24 using the ProLuCID search algorithm [25] for peptide identification. The search parameters were as follows: specific to trypsin with two missed cleavages, variable modification of methionine oxidation, fixed modification of carbamidomethyl cysteine, ± 10 ppm precursor-ion tolerance, ± 600 ppm fragment-ion tolerance, and ±10 reporter ion tolerance.

### 3.7. MRM-Based Targeted Proteomics Analysis for Purity

To select EV-specific marker proteins, we used our shotgun data and a list of the top 100 EV proteins released in the ExoCarta depository (http://exocarta.org/index.html, accessed on 10 April 2019) that were frequently detected as a result of various EV analyses including mass spectrometry. First, the top 100 EV proteins were compared with our proteome shotgun data representing evidence of expression in the EV fractions. Of them, the proteins common to both datasets were chosen. Second, the MRM methods were generated using an in-silico approach using Skyline 20.2 (MacCoss Lab Software, Seattle, WA, USA) [26]; upon further refinement of the selected peptides, the transitions were employed. At least three transitions from one proteotypic peptide were generated with 2 or 3 peptide charge states containing 8–30 amino acids. In addition, no post-translational modification (PTM) was applied, and non-specific cleavages were not allowed. Third, the MRM assays were experimentally refined using the EV fractions obtained from the pooled serum sample. Fourth, we determined the proteins selected in the third step using stable isotope dilution (SID)-MRM in non-purified and EV fractions. Then, we selected proteins that increased in the EV-purified fractions for purity evaluation.

We achieved MRM using a QTRAP 5500 hybrid triple quadrupole/linear ion trap mass spectrometer (Sciex, Framingham, MA, USA) equipped with an electrospray ion source. An MRM scan was performed in positive mode with an ion spray voltage of 5500 V. The MRM mode setting was as follows: curtain gas and spray gas were 30 and 50, respectively, and the collision gas was set to the medium level. The declustering potential (DP) was set to 57 V. The mass resolution was set to the units of the advanced MS parameter. The tryptic peptides were loaded into an InfinityLab poroshell 120 EC-C18 (Agilent, 2.1 × 50 mm) packed with C18 (2.7 µm, 120 Å) reverse-phase resin and separated by LC using a linear gradient of 5–40% buffer B for 6 min followed by 40–95% buffer B over 4 min at a flow rate of 0.4 mL/min.

### 3.8. Statistical Analysis

The acquired MRM assays were analyzed using MedCalc software version 19.0.7 (Mariakerke, Belgium). Differences in protein concentrations between purified and non-purified EV fractions were assessed by the nonparametric Wilcoxon test; *p* ≤ 0.05 was considered statistically significant.

## 4. Conclusions

This is the first report highlighting the usefulness of multiple-cycle polymeric EV precipitation and an MRM panel for purity evaluation. Here, we demonstrated the usefulness of this method through EV shotgun proteomics and EV protein quantification using an MRM assay. In particular, we demonstrated that a single polymeric reagent treatment is vulnerable to serum-derived contamination and does not provide sufficiently high purity for proteomics studies, based on our MRM assay results.

We were able to identify more EV-related proteins including plasma membrane proteins via multiple-cycle polymeric EV precipitation than those obtained using the conventional UC method. Even though further validation experiments are necessary in a large-batch clinical sample, our multiple-cycle polymer-based EV purification method with a high recovery rate using a small sample volume is expected to facilitate various clinical applications of EVs.

## Figures and Tables

**Figure 1 ijms-22-04311-f001:**
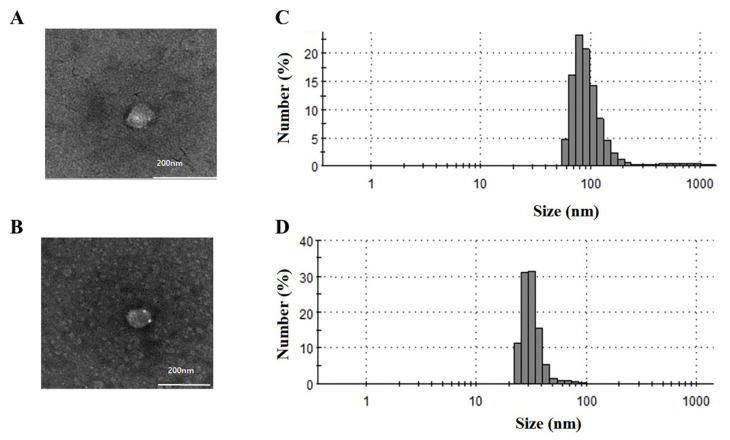
Characterization of EV fractions obtained from UC and multiple-cycle polymeric EV precipitation. Transmission electron microscopy (TEM) and dynamic light scattering (DLS) analyses were performed to determine the morphology of vesicles and particle size distribution in the (**A**,**C**) UC-EV and (**B**,**D**) 4-cycle EV fractions.

**Figure 2 ijms-22-04311-f002:**
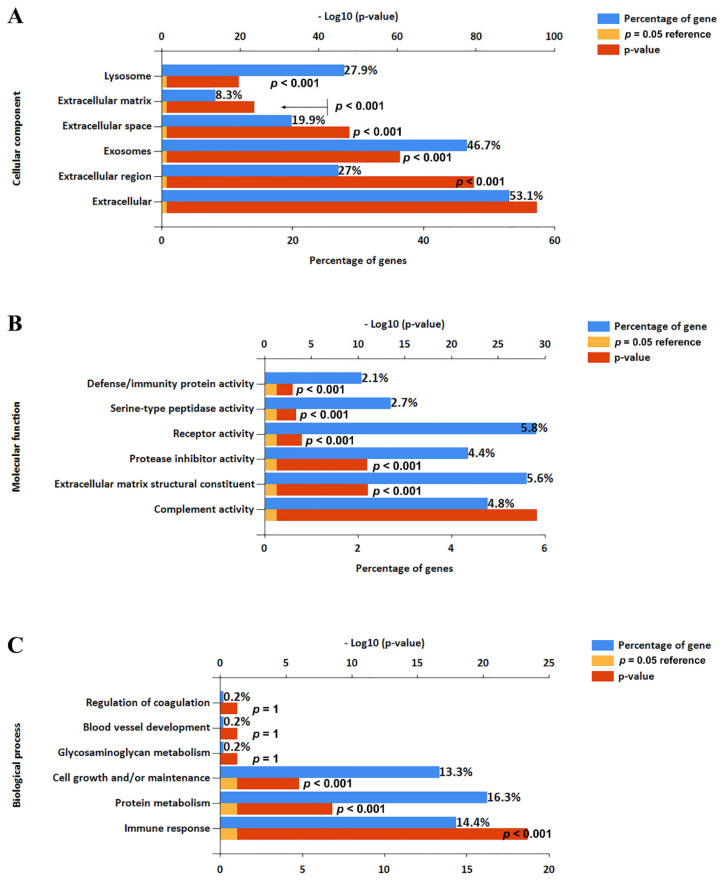
Functional analysis of identified proteins in 4-cycle EV fractions. Functional enrichment analysis was conducted using FunRich software: (**A**) cellular components, (**B**) molecular functions, and (**C**) biological process.

**Figure 3 ijms-22-04311-f003:**
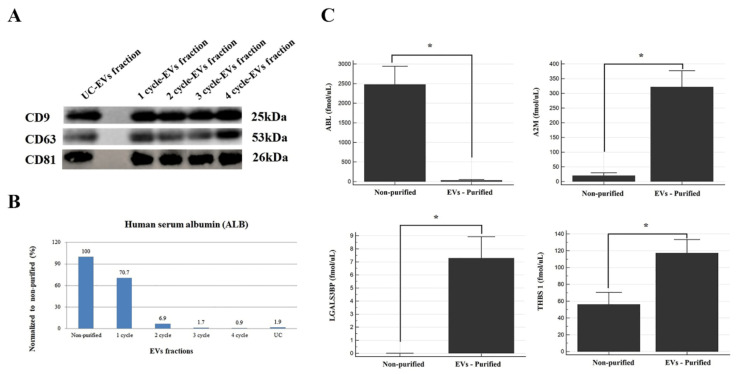
The purity evaluation. (**A**) Western blot analysis of CD9, CD63, and CD81 on 1–4-cycle EV fractions. (**B**) The ALB levels measured in EV fractions (1–4 cycles) and UC using an MRM assay. (**C**) The levels of 3 EV markers (A2M, LGALS3BP, and THBS 1) and ALB in non-purified and EV-purified fractions (2-cycle) obtained by an MRM assay. ***, *p*-value *<* 0.0001.

**Figure 4 ijms-22-04311-f004:**
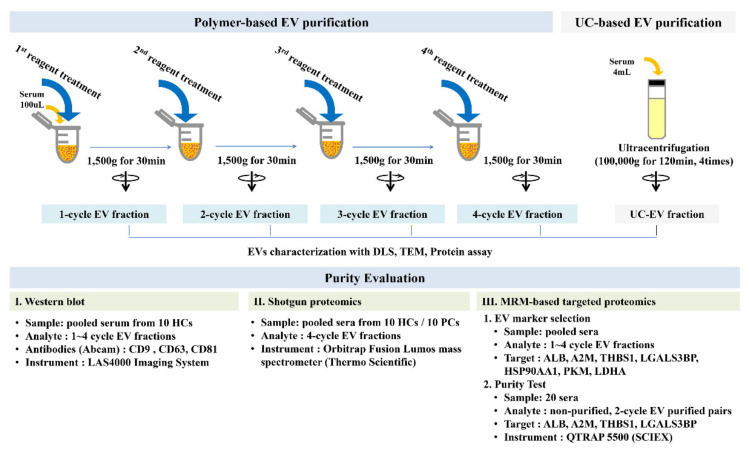
Experimental summary. The EV fraction was obtained by 1 to 4 cycles of polymeric reagent treatment and UC-EV after four repeated cycles of ultracentrifugation. DLS, TEM, and protein assays were performed on these EV fractions. EV proteins candidates were assessed in the 1–4-cycle EV fractions for MRM panel development for the purity test. For the purity test, we performed 2- and 4-cycle reagent treatment for MRM-based targeted proteomics analysis and shotgun proteomic analysis, respectively.

**Table 1 ijms-22-04311-t001:** The 27 CD markers identified in the 4-cycle EV fraction.

Uniprot ID	Gene Name	mcpEx	UC ^21^	Protein Evidence	Description	Spectral Counts	A Total Number of Unique Peptides
Q496F6	CD300E	v			CMRF35-like molecule 2	2	2
P15144	ANPEP	v	v	serum/exo/cyst	Aminopeptidase N	11	9
P21926	CD9	v	v		CD9 antigen	3	3
P02730	SLC4A1	v	v		Band 3 anion transport protein	5	5
P13987	CD59	v	v		CD59 glycoprotein	2	2
O14786	NRP1	v			Neuropilin-1	23	20
P11717	IGF2R	v			Cation-independent mannose-6-phosphate receptor	17	15
Q9NR16	CD163L1	v			Scavenger receptor cysteine-rich type 1 protein M160	3	3
P20963	CD247	v			T-cell surface glycoprotein CD3 zeta chain	3	2
P08637	FCGR3A	v			Low affinity immunoglobulin gamma Fc region receptor III-A	4	4
P05106	ITGB3	v	v		Integrin beta-3	12	10
P14151	SELL	v			L-selectin	2	2
P08571	CD14	v		serum/exo/cyst	Monocyte differentiation antigen CD14	2	2
Q86VB7	CD163	v			Scavenger receptor cysteine-rich type 1 protein M130	7	6
P15814	IGLL1	v			Immunoglobulin lambda-like polypeptide 1	10	5
P08514	ITGA2B	v	v		Integrin alpha-IIb	12	12
Q07954	LRP1	v	v		Prolow-density lipoprotein receptor-related protein 1	50	45
P14209	CD99	v			CD99 antigen	3	2
P12830	CDH1	v			Cadherin-1	2	2
P20702	ITGAX	v			Integrin alpha-X	3	3
P05556	ITGB1	v	v		Integrin beta-1	2	2
P05107	ITGB2	v			Integrin beta-2	2	2
P24071	FCAR	v			Immunoglobulin alpha Fc receptor	3	3
Q96RD9	FCRL5	v			Fc receptor-like protein 5	4	4
P20023	CR2	v			Complement receptor type 2	2	2
P28908	TNFRSF8	v			Tumor necrosis factor receptor superfamily member 8	3	3
P19320	VCAM1	v			Vascular cell adhesion protein 1	2	2

**Table 2 ijms-22-04311-t002:** MRM-MS of extracellular vesicles along with representative peptide signatures used to assess isolated EV purity.

HGNC Symbol	Peptide Sequence	Target Ion	Intact (m/z)		IS (m/z)		CE (eV)
			Q1	Q3	Q1	Q3	
ABL	FQNALLVR *	+2y6	480.8	685.4	485.8	695.4	26.2
		**+2y5**	**480.8**	**571.4**	**485.8**	**581.4**	**26.2**
		+2y4	480.8	500.4	485.8	510.4	26.2
THBS1	SITLFVQEDR *	+2y8	604.3	1007.5	609.3	1017.5	30.6
		**+2y6**	**604.3**	**793.4**	**609.3**	**803.4**	**30.6**
		+2y5	604.3	646.3	609.3	656.3	30.6
A2M	NEDSLVFVQTDK *	+2y10	697.8	1151.6	701.9	1159.6	34.0
		+2y7	697.8	836.5	701.9	844.5	34.0
		**+2y6**	**697.8**	**737.4**	**701.9**	**745.4**	**34.0**
LGALS3BP	YSSDYFQAPSDYR *	+2y9	799.8	1146.5	804.8	1156.5	37.7
		**+2y8**	**799.8**	**983.5**	**804.8**	**993.5**	**37.7**
		+2y7	799.8	836.4	804.8	846.4	37.7

* represents an amino acid labeled with a heavy isotope, ^13^C^15^N. Precursor/fragment ions in bold are used for quantification. The transitions (Q1/Q3) were optimized using Skyline software. Abbreviations: ABL, serum albumin; THBS 1, thrombospondin 1**;** A2M, alpha-2-macroglobulin; LGALS3BP, galectin 3 binding protein; CE, collision energy.

## Data Availability

The data presented in this study are available in the article or Appendix A.

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
