# Peer review of "Multiple-Cycle Polymeric Extracellular Vesicle Precipitation and Its Evaluation by Targeted Mass Spectrometry"

_ijms, 2021, doi:10.3390/ijms22094311_

Round 1
Reviewer 1 Report
In this manuscript the authors describe a novel technique for exosomes isolation by using polymeric based precipitation method.
Although the subject is scientifically important, especially for exosomes isolation from body fluids, there are several drawbacks.
Major issues:
- The authors should have compared their results to those of ultracentrifuge and the existing commercial kits (they have used ExoQuick) throughout their analysis.
- When introducing a novel method- the authors should at least try to explain the principle of action of their compound.
- To evaluate a novel method of exosomes isolation the authors should have also analyze the RNA content of the isolated exosomes.
- The figures are in a very bad resolution- one cannot read the written words- and therefore must be changed.
- In the conclusions section- there is a mix between the purity and the isolation- as I understand the main goal was to isolate exosome in a better way and the purity was the mean to evaluate is- (Line 258 for example).
Minor points:
- The source of the exosomes (human sera) should be stated at the beginning of the results and not only at the methods section.
- The statement in lines 86-88 is not clear.
- In the legend to Table 2- what is Q1/Q3?
- In line 234- there is a mistake (5 after 3…)
Author Response
Major issues:
1.The authors should have compared their results to those of ultracentrifuge and the existing commercial kits (they have used ExoQuick) throughout their analysis.
Answer: As pointed out, it would have been better to compare ultracentrifugation and other kits throughout our analysis, but this was not possible due to limited sample volume. The amount of EV recovery using UC is insufficient sample volume to carry out all the tests. Also, comparison of multiple commercial kits is not the purpose of our current study. We ask for your understanding of this point.
The strength of this study is provision of an easy protocol with enhanced recovery rate of high purity EVs without sample loss. We chose ExoQuick, the most commonly used commercial kit, and simply applied this reagent treatment in several cycles to isolate high-purity EVs. Then, the corresponding EV fractions were compared to those of UC by methods of WB, TEM, and DLS.
We showed that the total protein amount of EVs purified from our protocol was at least 4.6-fold higher than that of UC, even though the volume of the sample was 16 times smaller, without sample loss.
Moreover, the MRM assay of three proteins (A2M, THBS1, and LGALS3BP) showed significant increase after EV purification, while the ALB level was dramatically reduced up to 99% after EV purification. Our data demonstrate that multiple reagent treatment is excellent at removing serum-derived contaminant proteins and enriching EV-related proteins. In addition, we expect the MRM panel consisting of three EV proteins and ALB to be useful for assessing EV purity isolated by other techniques.
The above results suggest that the evaluation of the usefulness of the protocol we proposed is in line with the purpose of the study.
- When introducing a novel method- the authors should at least try to explain the principle of action of their compound.
Answer: This is a study optimizing a protocol to isolate high-purity EVs from small amounts of human serum using commercially readily available polymer-based reagents and introducing the MRM-MS assay method for purity evaluation. We added explanation about the need and importance of our study approach in the Introduction section (page 2, lines 64 to 72). If you let us know more specifically what it means to “explain the principle of action of their compound,” we will address this.
- To evaluate a novel method of exosomes isolation the authors should have also analyze the RNA content of the isolated exosomes.
Answer: This study is based on proteomics. We applied various methods (e.g., WB, protein assay, TEM, DLS) to evaluate the usefulness of the applied protocol (high purity, minimal sample loss) for proteomics study. Analyzing the RNA content is beyond the purpose and scope of this current study. Also, our laboratory does not conduct RNA research. We expect to be able to apply our method to future RNA studies.
- The figures are in a very bad resolution- one cannot read the written words- and therefore must be changed.
Answer: According to the reviewer's opinion, we modified all figures to a higher resolution. We followed the guidelines of the figure file format (minimum width/height 1000 pixels or resolution 300dpi or more) requested by the journal.
- In the conclusions section- there is a mix between the purity and the isolation- as I understand the main goal was to isolate exosome in a better way and the purity was the mean to evaluate is- (Line 258 for example).
Answer: Yes, the main goal was to isolate exosomes in a better way for future clinical application. However, the comprehensive purity assessment by targeted MRM-MS is another important part of our research. According to the reviewer's comment, we reviewed and clarified the Conclusion section.
Minor points:
- The source of the exosomes (human sera) should be stated at the beginning of the results and not only at the methods section.
Answer: We have corrected this sentence according to the reviewer's comment (page 2, line 80).
- The statement in lines 86-88 is not clear.
Answer: According to the reviewer's point of view, we have revised the sentences as follows.
- > We could not prepare enough EV fractions from UC experiments due to limitation of sample volume and very low recovery rate.
- In the legend to Table 2- what is Q1/Q3?
Answer: As you indicated, we have modified Table 2. Q1 and Q3 represent precursor ion and its fragment ion, respectively.
- In line 234- there is a mistake (5 after 3…)
Answer: As you indicated, we have corrected the sentence (page 9, line 242).
Reviewer 2 Report
Manuscript entitled "Multiple Cycle Polymer-Based Extracellular Vesicle Isolation and Its Evaluation by Targeted Mass Spectrometry" by Park and co-workers describes the aplicability of multiple-cycle polymeric extracellular vesicles precipitation in enhancement of the purity of the extracellular vesicles fractions with a small sample volume. The purity was analyzed using MRM method. The manuscript requires some changes to be published in the IJMS.
My comments are presented below. Give comments, make changes in the text.
- Introduction - provide information about the applicability of mass spectrometry in the presented study. Explain why did you decided to use MRM technique.
- Table 1 - give the m/z of parent ion, determine ion types ([M+H]+, others)
- Table 1 - explain clearly why only the presented peptides was applied in MRM analysis
- materials and methods - MRM based proteomics - determine parent ion, determine ion types ([M+H]+, others), determine fragment ions
- page 9, lines 244-245 - Agilent, 2.1 x 50 mm - use
× instead of x.
- check reference style
- make language corrections.
Author Response
- Introduction - provide information about the applicability of mass spectrometry in the presented study. Explain why did you decided to use MRM technique.
Answer: As you pointed out, we have added information related to the applicability of mass spectrometry including MRM technology in the Introduction section (page 2, lines 64 to 72).
-> A large sample size, poor yield of EVs, and the time-consuming process are major problems in clinical application of EV biomarkers. The evaluation of impurities from the EV purification process is critical for downstream EV proteomics analysis. On the other hand, evaluation of purity is limited to measurement of serum albumin using western blotting without EV-specific proteins. Development of an analysis method for purity evaluation is necessary. For quantitative analysis of newly discovered proteins, multiple reaction monitoring (MRM) is a more useful technique than other antibody-based methodologies (e.g., western blotting or enzyme-linked immunosorbent assay) due to its rapid development and clinical application.
- Table 1 - give the m/z of parent ion, determine ion types ([M+H]+, others)
Answer: As you pointed out, we have revised Table 2 (Table 1 changed to Table 2).
- Table 1 - explain clearly why only the presented peptides was applied in MRM analysis
Answer: As described in section 3.6. (MRM-based target proteomics analysis for purity), we used Skyline software to predict the tryptic peptide of the protein of interest and then selected peptides that met the selection criteria (e.g., 2 or 3 charge states, 8-30 amino acids, no post-translational modification (PTM), and uniqueness). The peptide with the highest response intensity was used for final quantification.
- materials and methods - MRM based proteomics - determine parent ion, determine ion types ([M+H]+, others), determine fragment ions
Answer: As you pointed out, we have revised Table 2 (Table 1 changed to Table 2) and added Supplementary Table S2 in the Material and Methods section.
- page 9, lines 244-245 - Agilent, 2.1 x 50 mm - use × instead of x.
Answer: As you indicated, we have revised this (page 9, lines 217 and 252).
- check reference style, make language corrections.
Answer: We checked all reference style and received English proofreading by “eWorldEditing” (https://www.eworldediting.com/).
Round 2
Reviewer 1 Report
The authors have related to the points that were raised earlier by me.
The only thing is the principle of the method; I would like them to write one sentence which explains this principle such as: our polymer binds specifically (?) to the ….. of the exosomes and in this way the obtained fraction is pure and concentrated… something like that.
Author Response
Answer: According to the reviewer's point of view, we have added the sentences in introduction session as follows.
- > Polymeric reagents reduce the solubility of EVs and allow less soluble components to precipitate out of solution. This method shows a high yield of EVs, but contamination of non-EVs proteins is a major problem. We expected that repeated treatment of polymeric reagents can wash away contaminated proteins and increase purity. Therefore, we proposed a modified method which includes multi-cycle processing to enhance the purity.
Reviewer 2 Report
Theo revised version od the presented manuscript meets all of my requirements.
Authors gave appropriate comments, made changes im the text and gave correct answers for
all of my questions. In the final version check spelling, correct languane.
Author Response
We appreciate the opportunity to revise our work for publication in International Journal of Molecular Science.